# Seasonal specialization drives divergent population dynamics in two closely related butterflies

Loke von Schmalensee [1,2] ✉, Pauline Caillault[1], Katrín Hulda Gunnarsdóttir[1], Karl Gotthard [1,2] & Philipp Lehmann[1,2,3]

Seasons impose different selection pressures on organisms through contrasting environmental conditions. How such seasonal evolutionary conflict is resolved in organisms whose lives span across seasons remains underexplored. Through field experiments, laboratory work, and citizen science data analyses, we investigate this question using two closely related butterflies (*Pieris rapae* and *P. napi*). Superficially, the two butterflies appear highly ecologically similar. Yet, the citizen science data reveal that their fitness is partitioned differently across seasons. *Pieris rapae* have higher population growth during the summer season but lower overwintering success than do *P. napi*. We show that these differences correspond to the physiology and behavior of the butterflies. *Pieris rapae* outperform *P. napi* at high temperatures in several growth season traits, reflected in microclimate choice by ovipositing wild females. Instead, *P. rapae* have higher winter mortality than do *P. napi*. We conclude that the difference in population dynamics between the two butterflies is driven by seasonal specialization, manifested as strategies that maximize gains during growth seasons and minimize harm during adverse seasons, respectively.

How species can coexist and biodiversity be maintained have for many decades remained fundamental questions in ecology[1–3]. Fields of study like life-history theory, and the debated niche and neutral theory, strive to understand the existence of the broad diversity of evolved strategies that organisms use to cope with environmental challenges[4–8]. A major source of such challenges is the substantial environmental variation associated with seasonality[9–14]. Yet, the concept of seasonality as a niche dimension, and how it imposes constraints on trait evolution, remains underexplored.

In seasonal environments, genotypes face the challenge of functioning over a wide range of environmental conditions. An important evolutionary solution to this challenge is phenotypic plasticity—the ability of one genotype to generate more than one phenotype[15–17]. For instance, organisms can adjust their life cycles to predictable seasonal conditions through migration or dormancy[12,18,19]. Still, if plasticity cannot fully compensate for the effects of seasonal changes, trade-offs arise when what is adaptive one season is maladaptive another[20–24]. Thus, seasonality can even maintain functional genetic variation within populations with short generation times through seasonally variable selection pressures. For example, recent studies on *Drosophila* reveal that rapid evolution can cause average phenotypes to track seasonal changes[20,25,26]. In organisms whose generation times instead span across seasons, multiple contrasting stable strategies are conceivable[27–30]. An organism that maximizes performance during the summer growth season at the cost of suffering greater losses during the adverse winter season could be considered a 'summer specialist'. Inversely, a 'winter specialist' might minimize harm during the winter season to achieve the same fitness.

[1]Department of Zoology, Stockholm University, SE-106 91 Stockholm, Sweden. [2]Bolin Centre for Climate Research, Stockholm University, SE-106 91 Stockholm, Sweden. [3]Department of Animal Physiology, Zoological Institute and Museum, University of Greifswald, 1D-17489 Greifswald, Germany. ✉e-mail: loke.von.schmalensee@zoologi.su.se

When selection pressures change throughout individuals' lives, it can lead to distinct evolutionary outcomes. This is perhaps clearest on spatial scales. For example, some salmon populations that grow together in the ocean but spawn separately in freshwater diverge most in their freshwater phenotypes, indicating local adaptation to temporary conditions mediated by phenotypic plasticity[31]. In contrast, migrating birds of the species *Junco hyemalis* express different phenotypes during winter depending on the geographic location of their breeding grounds, despite overwintering in sympatry[32]. Replacing seasons for locations makes it conceivable that, on a temporal axis, seasonality presents a range of environments that might allow species to evolve similar adaptations across many niche dimensions while diverging in their seasonal adaptations.

The general idea of seasonal niche divergence is not new. For instance, studies suggest that plants can avoid competition for pollinators by diverging in phenology (seasonal timing)[33–36], and sympatric *Rhagoletis* flies show evolutionary divergence associated with differences in host plant use, owing partly to asynchrony caused by different host plant phenologies[37–39]. Still, empirical studies have so far largely focused on the phenological aspect of the seasonal niche (with exceptions in the microbe and plankton literature[40–42], foreshadowed by Hutchinson's 'paradox of the plankton'[43]), that is, how separation in time of comparable life stages can promote coexistence[44]. Here, we instead focus on seasonal specialization—how differences in seasonal adaptations can cause divergent partitioning of fitness across seasons, such as in the aforementioned summer and winter specialists. In contrast to phenological differences, seasonal specialization can drive seasonal niche divergence even among sympatric and synchronous organisms that share resources, via seasonal changes with disparate effects on different genotypes[29,30,42,45–48].

We explore seasonal specialization in a study system of two closely related butterfly species, *Pieris rapae* and *P. napi*[49,50]. The two species co-occur in Sweden, geographically and phenologically (Fig. S1), and have seemingly similar life-history strategies. *Pieris rapae* and *P. napi* both overwinter as diapausing pupae, share voltinism (number of yearly generations) patterns across their sympatric Swedish range, lay individual eggs on cruciferous plants (Brassicaceae) and have equivalent larval performance on a range of host plants (Fig. S1)[51–53]. Additionally, the two species share host plant preferences under common garden conditions (though variation in host plant use is broader in *P. rapae*)[51,52]. Despite these ecological similarities, *P. rapae*

and *P. napi* display remarkably different population dynamics in their northern range (Fig. 1)[54].

*Pieris rapae* start the growth season with a relatively small first generation, but show drastic increases in population size in the second generation (Fig. 1). *Pieris napi*, on the other hand, display similar population sizes in both generations (Fig. 1). These patterns are consistent across years and geographic locations in Sweden (Fig. S1). Intuitively, this implies that *P. napi* are better at overwintering than *P. rapae*, and that *P. rapae* instead have higher rates of population increase during growth seasons, compensating for their winter losses. At northern latitudes with relatively short growth seasons—where *P. napi* go from being bivoltine to univoltine—observations of *P. rapae* cease. In other words, one single yearly (and thus overwintering) generation appears insufficient for maintaining a steady population of *P. rapae* (Fig. S1). Additionally, *P. rapae* and *P. napi* have been shown to differ in their host plant use in nature (but not in the laboratory), with *P. rapae* preferring plants in warmer and drier microclimates, in line with their destructive preference for warm and dry agricultural fields[52,55,56]. Recently, similar microclimatic differences in oviposition preference were demonstrated between Spanish *P. rapae* and *P. napi* populations and linked to experimental differences in heat mortality[57].

Despite (or maybe because of) *P. rapae*'s and *P. napi*'s ecological similarities, differences between the species have been widely documented[51,52,55,57–62]. Yet, *P. rapae*'s and *P. napi*'s remarkably divergent seasonal population dynamics have gone largely unnoticed, and the species' ecological and physiological differences have therefore never been linked to seasonality. Thus, we here put forth the hypothesis that differences in thermal adaptation are driving differences in seasonal population dynamics between *P. napi* and *P. rapae* in their northern range, where thermophilic *P. rapae* are favored during summer and *P. napi* are favored during winter. To investigate this, we compare *P. rapae* and *P. napi* through a combination of field experiments, laboratory experiments, and citizen science data analyses. We show that *P. rapae* do indeed prefer warmer microhabitats for oviposition during the growth season than do *P. napi*, even when host plants are standardized. Further, we show that these differences in thermal preference are linked to differences in thermal performance and tolerance, with *P. rapae* being consistently more thermophilic than *P. napi* in multiple growth season related traits. Instead, *P. napi* have higher overwintering success than do *P. rapae*. Linking field and laboratory results back to population dynamics, we show that *P. rapae* consistently produce more offspring that survive until adulthood during growth seasons compared to *P. napi*, but instead have higher mortality during winter. We conclude that thermal niche divergences can lead to differences in seasonal specialization, here represented by a strategy that maximizes gains during warm growth seasons and another that minimizes harm during adverse cold seasons.

## Results

### *Pieris rapae* prefer warmer microclimates for oviposition than do *P. napi*

We investigated whether microclimatic differences alone could explain the previously reported microhabitat separation between *Pieris rapae* and *P. napi*[52,55,57] by translocating standardized host plants (*Brassica napus napus*) to 36 temperature-monitored microhabitats in a field site with natural *P. rapae* and *P. napi* populations (Methods). Eggs laid on the translocated host plants were identified as either *P. rapae* or *P. napi*, and the species affiliation probability was modelled as a function of the average daily microclimate temperature using a logistic regression (Methods). The analysis revealed that *P. rapae* are more thermophilic than *P. napi* in their habitat choice: *P. rapae* are more likely to lay eggs in relatively warm microclimates, while *P. napi* prefer relatively cool ones (Fig. 2). These results align with previous findings[52,57], but importantly also distinguish temperature as a main predictor of oviposition habitat choice, with the effect of average daily microclimate temperature

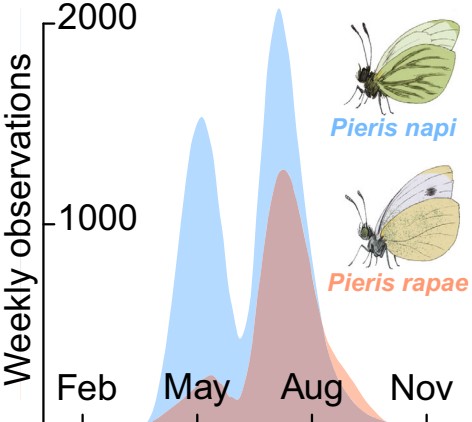

**Fig. 1 | Differences in within-year population dynamics.** Observations of *Pieris rapae* (red area *n* = 10,236) and *P. napi* (blue area, *n* = 20,492) within their common Swedish geographical range over the years 2010–2021. Both species display two distinct flight peaks, but in *P. rapae* the relative size of the spring peak, constituted by overwintering individuals, is much smaller than in *P. napi*. Data from the Swedish Species Information Centre[54].

showing a strong signal in the analysis (Fig. 2; slope: 0.67, CI$_{90}$: 0.33–1.08). Additionally, oviposition choice had a clear influence on the temperature that the offspring would have had experienced the following month; cold oviposition microclimates remained relatively cold and stable, whereas warm oviposition microclimates remained warmer on average but were highly variable (Fig. S7).

### *Pieris rapae* perform better in warm growth season conditions than do *P. napi*

To explore whether the observed differences in thermal preference between *P. rapae* and *P. napi* corresponded to differences in thermal physiology, we quantified growth season performance in development and growth (between 10 °C and 35–40 °C) for each separate life stage as well as the full ontogenetic development. We then fitted thermal performance curves (TPCs)[63,64] defined by four parameters ($T_{min}$, $T_{opt}$, $T_{max}$, and $R_{opt}$) to the development and growth rate data. We thereafter compared TPC parameter estimates between the species to infer differences in thermal adaptation. Additionally, we fitted log-linear regressions to pupal mass retention data (i.e. how much of the initial pupal mass were retained in newly eclosed adult butterflies), and investigated *P. rapae*'s and *P. napi*'s differences in thermal sensitivity of energetic development costs by comparing the slope estimates. For details, see Methods.

In all estimated TPCs, *P. rapae* showed a higher performance than *P. napi* at warm temperatures. In fact, with the exception of $T_{min}$ for larval growth rate, all separate TPC temperature parameter estimates ($T_{min}$, $T_{opt}$, $T_{max}$) in all life stages were higher for *P. rapae* than *P. napi*. This signifies that *P. rapae* thermal reaction norms are right-shifted relative to those of *P. napi* (Fig. 3a–e). Although $T_{max}$ estimates are inherently associated with high model uncertainty (see Appendix S1) we emphasize that the species differences are consistent across independent data and models (as visualized in Fig. S5). Moreover, in accordance with the 'warmer is better' hypothesis[63,65], *P. rapae*'s higher $T_{opt}$ is associated with higher $R_{opt}$ (Fig. 3a–e). For each specific TPC parameter estimate, see Supplementary Data 1.

In *P. napi*, increased temperatures had a clear negative effect on pupal mass retention, leading to smaller adults relative to the initial mass at pupation (Fig. 3f; slope: −0.0083, CI$_{90}$: −0.014 to −0.0037). However, in *P. rapae*, slope estimates were relatively flat (Fig. 3f; slope −0.0019, CI$_{90}$: −0.0061 to 0.0031). This lower thermal sensitivity in *P. rapae*'s ability to retain mass throughout pupal development indicates that they experience lower relative energetic costs when developing at warm temperatures than do *P. napi*.

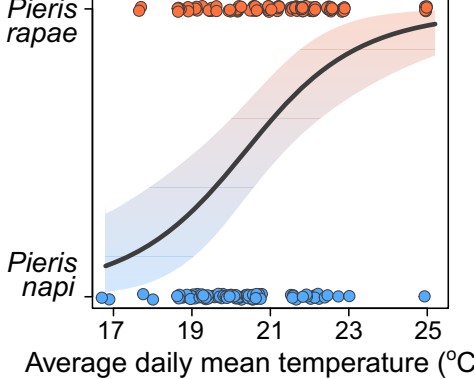

**Fig. 2 | Differences in temperature preference for oviposition.** Logistic regression describing how the species affiliation probability of an egg laid on *Brassica napus napus* depends on the microclimatic daily mean temperature during the second flight peak. Points represent individual eggs and the band shows the 90% credible interval, colored by species probability. Microclimate data are previously published[135].

### *Pieris rapae* survive heat, *P. napi* survive winter

We recorded individuals that had completed development in each treatment as survivors. Temperature- and species-specific survival probabilities during development were subsequently modelled using a logistic model (Methods). Additionally, we reared diapausing pupae (12 families from each species) that were kept under simulated winter conditions until they were moved to 17 °C (permissible for development) after ~7 months (Methods; Fig. S9). Individuals that successfully eclosed as adult butterflies were recorded as winter survivors.

*Pieris rapae* and *P. napi* differed in thermal tolerance at high developmental temperatures (Fig. 4a). At 32 °C the estimated average probability of *P. rapae* surviving a life stage was estimated to 75% (CI$_{90}$: 50–89%) whereas the same estimate for P. napi was 38% (CI$_{90}$: 12–66%). At 35 °C, none of the 192 *P. napi* individuals survived development, whereas some *P. rapae* completed development in all life stages (Fig. 3a–e and Fig. 4a). The average probability of *P. rapae* survival at 35 °C was estimated to 8% (CI$_{90}$: 3–23%). For specific estimates of developmental survival probabilities in all treatments, see Fig. 4a and Table S1.

In overwintering survival, the direction of difference between the species was reversed. *Pieris rapae* had a lower average survival during winter diapause (estimate: 73%, CI$_{90}$: 64–83%) than had *P. napi* (estimate: 89%, CI$_{90}$: 83–94%). Notably, two dead *P. rapae* individuals had also resumed development under cold conditions, even though they were previously in developmental arrest, indicating a shallower developmental suppression than in *P. napi*.

### *Pieris rapae* and *P. napi* consistently partition fitness differently across seasons

We gathered observational citizen science data of adult *P. rapae* and *P. napi* (2010–2021) from the Swedish Species Information Centre[54]. We structured the observations temporally and spatially by grouping them by year and province, only including provinces where both species were observed consistently (Fig. S1). For each species and group, we extracted peaks in observation numbers numerically through kernel density estimation. The peaks represented distinct generations in our analyses, and relative generation sizes were approximated by multiplying the density of the peaks by the total number of observations in their respective groups. For subsequent analyses, we removed a small minority of groups without two detectable distinct generations ('spring' and 'fall'). We modelled log-transformed generation sizes as linear functions of the log-transformed previous generation sizes for both species (accounting for the temporal and spatial structure of the data). This was done separately for spring and fall generations, resulting in two models, comparing *P. rapae* and *P. napi* population growth and decline across summer and winter, respectively. For details, see Methods.

The models revealed that spring peak generation size was a main driver of fall peak generation size within years and provinces (Fig. 5a). In *P. rapae*, the slope was slightly flatter (slope: 0.65, CI$_{90}$: 0.53–0.78) than in *P. napi*, (slope: 0.8, CI$_{90}$: 0.68–0.89). A flatter slope represents in the model that each additional individual in one generation contributes less to the size of the following generation, which indicates that *P. rapae* are subjected to stronger negative density dependence over the growth season than are *P. napi*. However, we caution over-interpretation of these slope differences since they might partly arise from 'regression to the mean'-effects (i.e., very small first generations are more likely to by chance lead to large growth estimates). Still, the higher intercept for *P. rapae* (Fig. 5a; intercept: 0.77, CI$_{90}$: 0.48–1.0) than for *P. napi* (Fig. 5a; intercept: 0.21, CI$_{90}$: 0.058–0.35) means that for all realistic scenarios, a single *P. rapae* adult in the spring peak will produce more offspring that survive until adulthood than will a single *P. napi* adult (Fig. 5a).

In the spring peak generation sizes, more of the total variation was attributed to the effects of year and province, as indicated by the

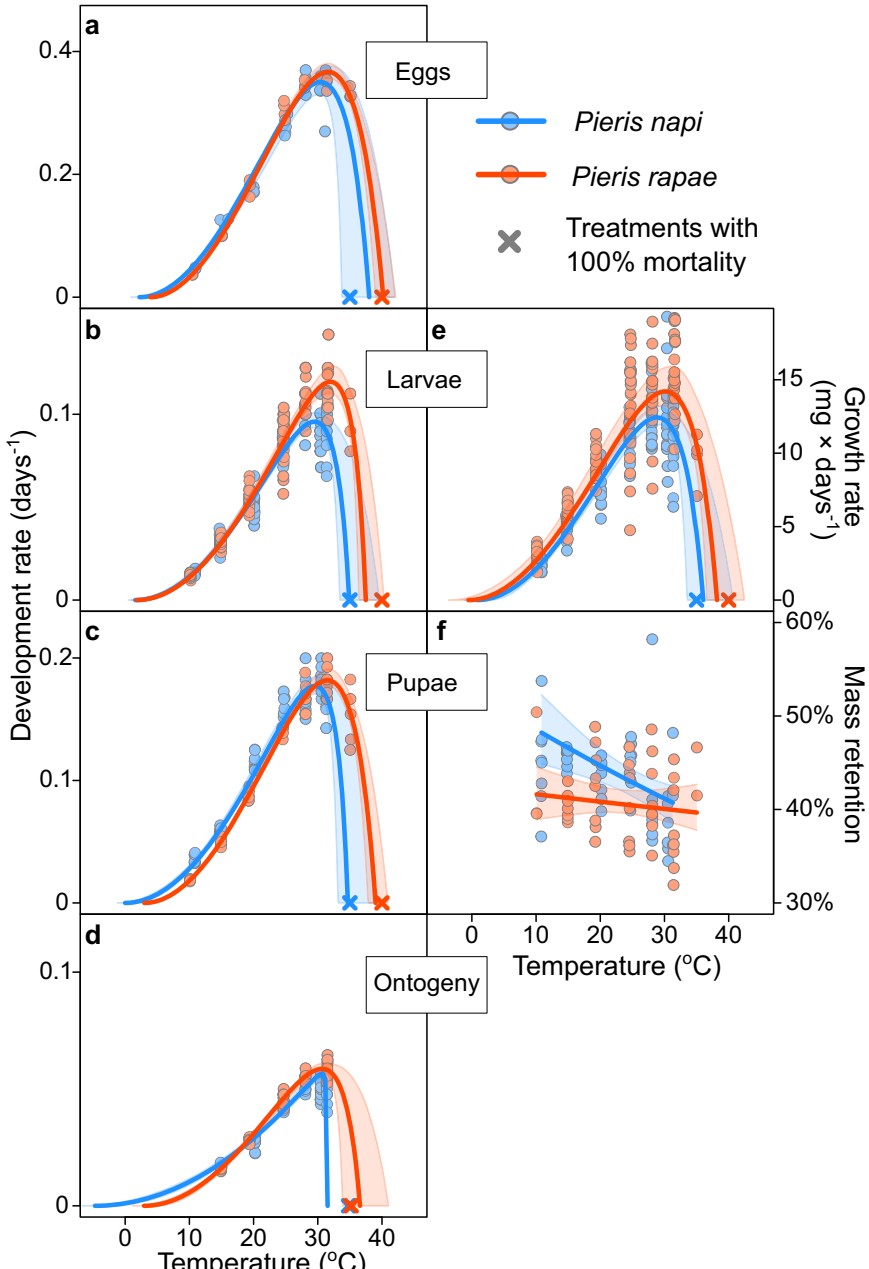

**Fig. 3 | Differences in thermal performance.** Estimated effects of temperature on growth season traits. **a–d** Temperature-dependence of development rates for each separate life stage (eggs, larvae, and pupae), and the full ontogenetic development (from oviposition to eclosion). **e** Temperature-dependence of larval growth rates. **f** Temperature-dependence of pupal mass retention (the percentage of mass at pupation that is retained as a newly eclosed adult butterfly). Points represent group means (e.g., each unique combination of sex, family, and treatment), lines represent estimated TPCs (based on posterior modes) and bands show 90% credible intervals. Egg and larval development data for *Pieris napi* are previously published[60].

relatively flat slopes (Fig. 5b). This is potentially because pupal diapause during the overwintering period is substantially longer than the pupal development during the growth season, leading to prolonged exposures to external conditions and increased chances of significant random events. Still, a general effect of previous fall generation sizes on the sizes of consecutive spring generations was detected with a high degree of confidence (Fig. 5b; slope: 0.26, CI$_{90}$: 0.14–0.37; no support for species-specific slopes). The intercept estimate for *Pieris rapae* (Fig. 5b; intercept: −1.8, CI$_{90}$: −2.1 to −1.5) was lower than for *P. napi* (Fig. 5b; intercept: −0.19, CI$_{90}$: −0.50 to 0.11), meaning that, while populations decline over winter in both species, a single *P. napi* adult in the fall generation will on average produce more offspring that survive until adulthood than will a single *P. rapae* adult. This difference is likely

driven by differences in pupal overwintering success, since the majority of the time between growth seasons is spent as overwintering pupae in both species.

While the absolute number of butterfly observations within a single year and province is not on its own informative of a single species' fitness, relative differences over time can be. Here, when confounding effects are accounted for, the effect of the abundance in one peak on the abundance of the next is a proxy for the average fitness over that time period. Thus, our models estimate that for a given population size, *P. rapae* have on average 1.8–2.9 times higher fitness than *P. napi* over summer, but *P. napi* instead have on average 5 times higher fitness than *P. rapae* over winter (Fig. 4). Note that, although *P. rapae* are generally more dispersive than *P. napi*[52,66], the

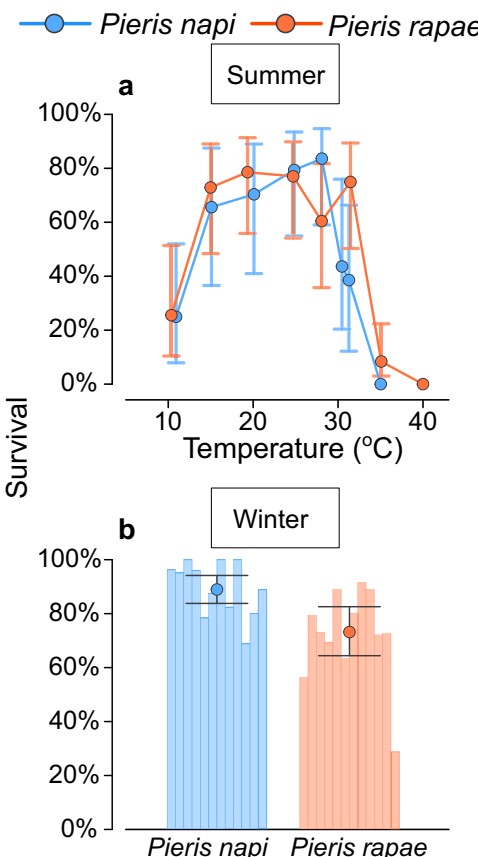

**Fig. 4 | Differences in overwintering success.** Estimated survival during development and overwintering in diapause. **a** Estimated relationships between constant temperatures and average survival during development (sample sizes from coldest to warmest treatment were $n = 170$, $n = 159$, $n = 184$, $n = 222$, $n = 198$, $n = 191$, $n = 255$, $n = 157$ *Pieris rapae*, and $n = 204$, $n = 173$, $n = 180$, $n = 217$, $n = 215$, $n = 291$, $n = 297$, $n = 192$ *P. napi*). **b** Overwintering success in diapausing individuals (individual bars represent families; sample sizes were $n = 652$ *P. rapae*, $n = 251$ *P. napi*). Points represent survival point estimates (posterior modes), whiskers represent 90% credible intervals.

clear signals across generations within locations and the timing of the *P. rapae* flight peaks across the Swedish range (Fig. S1) strongly imply that the peaks are formed mainly by local individuals, not migrants.

### *Pieris rapae* population sizes are catching up with those of *P. napi*

We performed a Poisson regression to model yearly changes in the number of butterfly observations across Sweden (2010–2021), accounting for the spatial structure in the data by fitting province-specific random intercepts. The model revealed that the relative number of *P. rapae* observations over 11 years (Fig. 6a; slope: 0.087, $CI_{90}$: 0.082–0.092) has increased at a significantly higher rate than the relative number of *P. napi* observations (Fig. 6a; slope: 0.034, $CI_{90}$: 0.031–0.037), suggesting that *P. rapae* populations are growing relative to *P. napi* populations (Fig. 6a–b).

### Differences between *P. rapae* and *P. napi* are concordant across analyses

A multitude of independent (e.g. life stage specific) and near-independent (e.g. $T_{min}$ and $T_{max}$ within a model) estimates point in the same direction: *P. rapae* are more thermophilic than *P. napi*, and therefore generally perform better in growth season-related traits, particularly at warm temperatures. On the other hand, *P. napi* perform

better during winter. Only one out of 23 parameters suggested *P. napi* to be more thermophilic than *P. rapae* in some area: $T_{min}$ for larval growth rate (it was estimated to be 1.3 °C lower for *P. rapae*). The concordance among all the results is demonstrated in Fig. S5. All parameter estimates and their associated uncertainties are detailed in Supplementary Data 1, including sex, year, and group-level effects. Finally, although we have not here focused on TPC differences among life stages it is an interesting research topic for which we refer to the results presented in Supplementary Data 1 (and we encourage new analyses of our raw data tailored for that specific purpose).

### Discussion

Our results indicate that a thermal niche divergence acts in concert with seasonal variation to create substantial differences in population dynamics between *Pieris rapae* and *P. napi*. First, *P. rapae* prefer warmer microclimates for oviposition in nature than do *P. napi* (Fig. 2). Second, *P. rapae* are consistently more thermophilic in trait performance and survival during development than are *P. napi* (Fig. 3, Fig. 4a), suggesting that the differences in microclimate preference reflect females tracking optimal environments for offspring growth and development (note that optimal mean temperatures are lower than $T_{opt}$[67]). This conclusion was strengthened by quantitative predictions of larval growth and development in each microhabitat during the month following oviposition, where increased oviposition temperatures correlated with increased relative performance differences in *P. rapae's* favor (Appendix S1, Fig. S8). Third, *P. rapae* have lower overwintering success under experimental conditions than do *P. napi* (Fig. 4b). Fourth, citizen science data analyses support the experimental results, showing that *P. rapae* in nature consistently have higher population growth during summer but lower overwintering success than do *P. napi* (Fig. 5). Relative to *P. napi*, *P. rapae* appears to be a 'summer specialist', capitalizing on warm summer conditions at the expense of overwintering success (Fig. 6c).

It is not yet clear how and why these differences between *P. rapae* and *P. napi* are maintained. Particularly, the low overwintering success of *P. rapae* in nature appears maladaptive (Figs. 1, 5b, 6b). We propose three non-exclusive explanations for this. First, it is possible that *P. rapae's* apparent maladaptation to northern winters is upheld by gene flow ('gene swamping') from southern populations that experience milder winters; *P. rapae* are generally more dispersive than *P. napi*, and a weaker spatial population structure likely hinders local adaptation[52,66,68–71]. Second, Swedish *P. rapae* populations might be young and mid-adaptation so that current genotypes do not accurately reflect the present selective environment[72–75]. If *P. rapae* historically evolved under warm conditions, current maladaptive winter responses in their Northern range can simply be an unresolved problem of selection past. The recent *P. rapae* abundance increases in Sweden (Fig. 6a, b) lend some support to this idea (populations have not yet stabilized). Additionally, *P. rapae* were reported to be neither very widespread nor common in Sweden in 1955[76], which is arguably not true today. Third, adaptive potential could be constrained by cross-seasonal trade-offs, that is, adaptations to winter conditions might come at the cost of maladaptations to summer conditions. We find some empirical support for this: there is heritable variation for overwintering success in the laboratory (Fig. 4b, Supplementary Data 1), and wild *P. rapae* populations have consistently been decimated over 11 winters (Figs. 5b and 6b), implying that natural selection should favor better overwinterers. Thus, without trade-off constraints, it appears plausible that higher overwintering survival would have evolved in the northern *P. rapae* populations (though this could be counteracted by gene flow).

Causally linking the above explanations to the observed trait correlations in *P. rapae* and *P. napi* is not trivial[77–82]. For example, many underlying mechanisms could lead to cross-seasonal trade-offs. Specialist–generalist trade-offs can arise through pleiotropy when

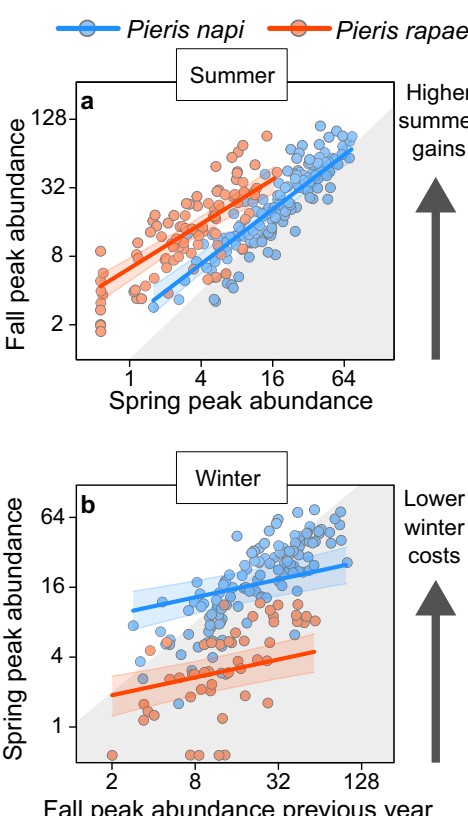

**Fig. 5 | Differences in summer and winter success in nature.** Estimated relationships between the abundance of two consecutive flight peaks. **a** The relationship between the abundance in the spring peak of a given year and province (*x*-axis) and the abundance of the following fall peak that year in the same province (*y*-axis). **b** The relationship between the abundance in the fall peak of a given year and province (*x*-axis) and the abundance of the following spring peak the next year in the same province (*y*-axis). Grey shaded areas represent estimated population declines between the two peaks, white areas represent estimated population increases. A slope of 1 (parallel to the intersection between the white and grey area) would indicate an absence of density dependent effects. Slope estimates less than 1, as observed here, represent negative density dependence, that is, individual reproductive success decreases with abundance (e.g. through competition, or density dependent parasitism or predation). Arrows show the general effect of an intercept change in an upwards direction. Points represent data from single provinces and years, and lines represent the estimated effect of the abundance in one flight peak on the abundance in the following (based on posterior modes), after accounting for random effects. Bands around the lines represent 90% credible intervals. Note the logarithmic scale of the axes. For visualization purposes, the values on the axes have been replaced with arbitrary values proportional to those used in the model.

genes have similar functions across different seasons[15,83–90]. Even when such seasonal conflict is resolved through plastic responses (e.g. diapause), allocation trade-offs can arise when increased efficacy in some traits (e.g. cold tolerance or developmental suppression) comes at the cost of decreased efficacy in others (e.g. heat tolerance or fecundity)[91–98]. It is plausible that gene swamping, lagging adaptation, and cross-seasonal trade-offs all play different roles in maintaining the seasonal differences between *P. rapae* and *P. napi* in nature, and we note that these processes remain important and underexplored topics for future studies on seasonal adaptation.

Whatever evolutionary processes maintain the seasonal differences between the two butterfly species, *P. rapae*'s superior summer performance (Fig. 5a) is driven by higher development rates (Fig. 3a–d), growth rates (Fig. 3e), survival[57] (Figs. 3 and 4a), and fecundity[51,52,58] at warm temperatures. The reasons for *P. rapae*'s

inferior overwintering success (Fig. 5b) is less clear, but possible explanations include poorer cold tolerance or metabolic suppression (leading to resource depletion)[99] causing higher winter mortality, or a less reliable developmental arrest during diapause, causing premature winter eclosions. We find support for both: experimental over-wintering mortality was higher in *P. rapae* than in *P. napi* (Fig. 4b), and two diapausing *P. rapae* individuals prematurely eclosed during the cold winter treatment. Such shallow winter diapause does not occur in *P. napi*[100] and could result from co-option of summer diapause mechanisms (seen in several other thermophilic Pierid butterflies[101–103]), potentially making *P. rapae* particularly sensitive to brief winter warm spells[104]. Experimental differences in overwintering success between *P. rapae* and *P. napi* are smaller than in those observed in nature (cf. Figs. 4b and 5b), but this is concordant with our expectations, since wild butterflies occasionally experience extreme cold and warm periods (Fig. S9).

To put our results in a broader context, it is important to consider that the thermal niche is linked to not only temperature but also other selective agents. For example, incidental grazing selects for close-ground oviposition in *Euphydryas editha* butterflies, subjecting off-spring to more extreme heat[105]. Inversely, thermal adaptations can open up for exploitation of new environmental resources, such as agricultural crops. *Pieris rapae* is a major agricultural pest, and the open fields of cultivated crucifers where *P. rapae* often fly and oviposit tend to be dry and warm[51,56–58,106,107]. Yet, *P. rapae* prefer dry and warm climatic conditions even outside agricultural fields, while *P. napi* that avoid such conditions also to a larger extent avoid agriculture (Fig. 2)[56]. Likely, *P. rapae*'s preference for, and tolerance to, high temperatures were pre-requisites for their heavy exploitation of Brassicaceae crops. In a broader sense, thermophilic insects are likely pre-adapted to anthropogenic land use due to its impacts on microclimate[108]. It is even possible that agriculture is necessary to sustain *P. rapae* populations in temperate areas, compensating for harsh winters by providing abundant food during growth seasons.

Understanding seasonal specialization can also help predict responses to a warming world. For instance, over the last 11 years, the summer specialist, *P. rapae*, has been increasingly favored relative to the winter specialist, *P. napi*; across Sweden *P. rapae* population sizes are catching up to those of *P. napi* (Fig. 6). Pinpointing exact reasons for this is difficult since insect responses to climate warming can be highly complex, even in the absence of evolutionary responses[109]. However, our findings underline the importance of considering the full seasonal cycle when making predictions about organismal responses to climate warming in the future, particularly since the climate changes disproportionally across seasons[14,110–113]. For example, increasing proportions of dry and warm microhabitats during the growth season[114,115] could favor *P. rapae* over *P. napi* in their sympatric range. On the other hand, by choosing cooler microclimates, *P. napi* could have more flexibility in coping with increasingly narrow upper thermal safety margins during growth seasons in the future[116]. Warming winters can benefit *P. rapae* if their overwintering success is primarily limited by cold tolerance, reducing the frequency of extreme cold events[117], but could also increase occurrences of premature *P. rapae* eclosions if their diapause is poorly regulated. For *P. napi*, warming winters are most likely detrimental, since they can delay diapause termination[100] and increase metabolic costs[118]. The latter effect has dual causes, since warmth both prolongs diapause and increases metabolic rates[119–122]. All taken together with the current climate trajectory, it appears likely that *P. rapae* will benefit in comparison to *P. napi* in the future.

While *P. rapae* might displace *P. napi* in their sympatric range (Fig. 6), northern areas where *P. napi* are univoltine do not sustain *P. rapae* populations (Fig. S1). This is likely because of constraints imposed by *P. rapae*'s seasonal specialization. Each winter, Swedish *P. rapae* populations are decimated (Figs. 5b, 6b), but each growth season, the same *P. rapae* populations grow drastically, compensating for

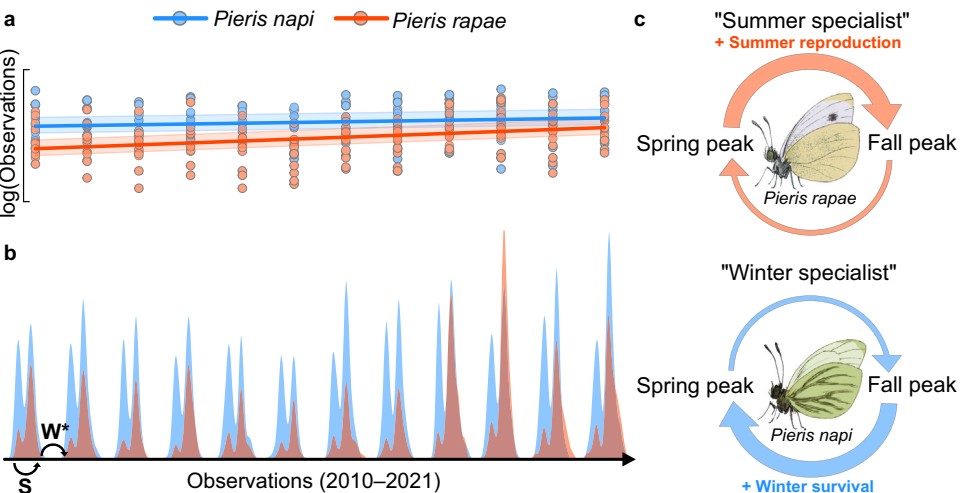

**Fig. 6 | The summer specialist is catching up to the winter specialist. a** Estimated proportional increases in butterfly observations over 11 years for *Pieris rapae* and *P. napi*. Note the steeper slope for *P. rapae*, indicating that its population sizes are increasing relative to *P. napi* population sizes. Points are clustered on the *x*-axis by year (2010–2021) and represent total observations per province that year. Lines show the estimated increase of observations over the years (based on posterior modes), and bands represent 90% credible intervals. **b** Density plot showing the distribution of *P. rapae* (red areas) and *P. napi* (blue areas) observations across their common range in Sweden over the years 2010–2021. Peak amplitudes have been scaled by the total number of observations for each species that year. Since observations span relatively large latitudinal gradients, the phenology has for visualization purposes been centered within each province. The letter S denotes one summer period, and the letter W denotes one winter period. The relative length of the winter period has been shortened to save space. **c** Conceptual demonstration of *P. rapae*'s and *P. napi*'s seasonal specialization. Upper arrows represent summer, lower arrows represent winter, and large arrows represent high relative fitness.

the winter losses (Figs. 5a, 6b). Such a strategy relies on (at least) one extra generation during the growth season. Without it, population sizes would be consecutively reduced each generation. Unable to recover during summer, *P. rapae* would quickly face extinction in absence of evolutionary rescue[123]. Yet, if adaptations responsible for the success of *P. rapae* in their northernmost bivoltine range impose constraints on *P. rapae*'s ability to become univoltine, the evolution of a viable univoltine strategy is rendered unlikely by gene swamping[68]. General empirical support for gene swamping is rare[124], but the present system might be an interesting exception to this pattern for the following reasons: (1) *P. rapae* populations are thriving at their northern bivoltine range margin, constituting an abundant source of genotypes that are maladapted to univoltinism (Fig. 6, Fig. S1); (2) *P. rapae* are strong dispersers[52,66,68–71]; (3) the transition to univoltinism is discrete, creating a particularly steep selection gradient; (4) the summer specialist phenotype of *P. rapae* is associated with multiple thermal traits (Figs. 2–4 and 6c), and is likely highly polygenic. Thus, using fitness landscape terminology[125], *P. rapae* populations must seemingly pass through a deep fitness valley to evolve univoltinism north of their present range margin. Consequently, northern (and high-elevation) habitats might constitute important refugia for *P. napi* and other winter specialists in the future.

In summary, we have demonstrated how two superficially similar butterflies differ drastically in their season-specific success. We propose that when season-specific success is constrained (e.g. by trade-offs), organisms whose lives span across seasons can evolve multiple stable and viable strategies that differ in seasonal specialization. For example, local fitness optima can in temperate environments be reached by either maximizing population growth during growth seasons or minimizing harm during adverse seasons. Although the underlying mechanisms might be fundamentally similar, this evolutionary outcome differs from that in more short-lived organisms such as *Drosophila*[126], which can adaptively track the seasonal environment through rapid evolution[25].

The idea of seasonal specialization has broad implications. For example, the 'warmer is better' hypothesis[63,65], positing that warm-adapted insect species generally have higher maximum population

growth rates than do cold-adapted ones, has focused on growth season performance (direct development in the laboratory). Whether this hypothesis holds true over the full seasonal cycle in temperate regions remains unknown, since warm-adapted insects might perform worse during winter. We emphasize that seasonality is a major source of temporal variation in selection that can introduce evolutionary conflict, for example among life-stages. Seasonality is a high-level dimension in niche space that captures substantial variation in many other lower-level niche dimensions, leading to suites of seasonal traits that evolve together. Consequently, seasonality must be carefully considered when investigating thermal adaptation in temperate regions[14,127–133]. As much else, this is of particular importance in light of the rapid climatic changes the earth has faced in the recent past, and the impending changes that await[114,134].

## Methods

### Oviposition choice (field experiment)

In 2019, rapeseed plants (*Brassica napus napus*) were grown in a common greenhouse and subsequently translocated to 36 microhabitats across a field site in Södermanland, Sweden (58°58′23.2″N, 17°09′19.7″E)[135]. Sites were selected manually from a total 110 sites[135] to ensure accessibility during the experiment, while capturing as much as possible of the variation in mean temperatures, and variation around those mean temperatures (based on temperatures the previous summer). All plants were translocated simultaneously (within an hour), two days after the cotyledons had sprouted, to standardize the quality of the plants (*B. napus napus* cotyledons are attractive to ovipositing females[56,136]). Plants were watered daily. The experiment was carried out between 19th and 25th of July, during the peak of the second adult generation (for simplicity referred to as the 'fall peak', in contrast to the 'spring peak'). To ensure that plant quality remained high throughout the experiment, two batches of plants were translocated successively, each for three days. From each batch, four pots with rapeseed plants were placed in all microhabitats (Fig. S2). Temperatures were measured hourly at each microhabitat, using shaded loggers with internal temperature sensors placed next to the plants (EL USB-1; Lascar Electronics, Salisbury,

UK; mounted in horizontal white PVC tubes, 30 cm long, 7.5 cm ø, 10 cm above ground; Fig. S2). Microclimate data are previously published[135]. After oviposition by wild butterflies, plants from the same sites were moved to the laboratory in separate plastic containers with wet paper towels in the bottom to maintain high humidity. Individuals were reared on *B. napus napus*, and surviving pupae were identified as either *Pieris rapae* or *P. napi*.

## Growth season traits (laboratory experiments)

In 2018 and 2019, mated *P. napi* and *P. rapae* females were collected in the Stockholm area, Sweden (WGS84 decimal: Lat. 59.368, Lon. 18.061). Individuals from the F2 generation were treated at multiple constant temperatures under long-day light conditions (23L:1D). Temperature treatments spanned different portions of development—both separate developmental life stages (eggs, larvae, and pupae) and the full ontogenetic development from oviposition to pupal eclosion (hereby 'ontogeny'). For the larval and pupal treatments, individuals had been reared under ambient conditions (22L:2D, 23 °C) before treatment was initiated. The measured developmental variables were development time (days to completion) and survival (yes/no). For larvae and pupae, mass at pupation were recorded. Additionally, the masses of newly eclosed adults were recorded in the pupal treatment. Sex was determined when possible, excluding the egg treatment and individuals that died before pupation since individuals were sexed as pupae.

In 2018, life stage-specific development of *P. napi* individuals from five families was measured at six constant temperature treatments (10, 15, 20, 25, 30, 35 °C). In 2019, two additional treatments (28, 32 °C) were added to increase sampling resolution, since no *P. napi* individuals survived development at 35 °C. This yielded in total eight *P. napi* temperature treatments. Sample sizes ranged from 39–96 per life stage and treatment ($n_{tot}$ = 1669). Termaks KBP 6395-L (Termaks, Bergen, Norway) climate chambers were used for the 10, 20, 28, 30 and 32 °C treatments, and Panasonic climate chambers (Panasonic MLR-352, PHC Europe B.V., Etten-Leur, Netherlands) were used for the remaining 15, 25, and 35 °C treatments.

The experiment was replicated with *P. rapae* in 2019. Individuals from four families were treated at seven different constant temperatures (10, 15, 20, 25, 28, 32, 35 °C). Sample sizes ranged from 13 to 102 individuals per life stage and treatment ($n_{tot}$ = 1539). Because pupae can easily be kept in individual containers, and do not move or feed, they are less susceptible to effects from other external factors, which reduces statistical noise in the data. Therefore, sample sizes were lowest in the pupal treatments to optimize the use of available individuals. While *P. napi* had 0% survival at 35 °C, some *P. rapae* individuals completed development at 35 °C in all except the ontogeny treatments. Therefore, spare individuals were used to investigate *P. rapae* survival in a 40 °C treatment. Sample sizes were dependent on the availability of individuals at the right developmental stage – 134 for eggs, 18 for larvae, and 4 for pupae. No individuals survived at 40 °C. If a single surviving *P. rapae* pupa had been recorded at 40 °C because of an increased sample size, survival would have been estimated to >0%, which further would have strengthened our conclusions about heat tolerance differences between *P. rapae* and *P. napi*. Thus, our conclusions are robust even considering the low power of the 40 °C pupal treatment. A Panasonic climate chamber was used for the 40 °C treatment and Termaks climate chambers were used for the other treatments.

The egg and larval development data for *P. napi* have previously been published, where the rearing procedure (which also applies to the *P. rapae* experiments) is outlined in detail[60]. Actual temperatures were measured hourly in each climate cabinet using HOBO MX2202 loggers (Onset Computer Corporation, Bourne, Massachusetts, United States). The measured temperatures were used for the subsequent modelling.

## Overwintering (laboratory experiment)

In 2019, diapausing *P. napi* pupae (251 F2 individuals from 12 families) and diapausing *P. rapae* pupae (652 F2 individuals from 12 families) were reared and overwintered under laboratory conditions. The pupae first remained at 23 °C for two weeks. They were then moved to 17 °C and after two additional weeks to 8 °C. After two months at 8 °C, they were moved to 2 °C where they remained for 5 months. The stepwise decrease in temperature was performed to allow pupae to acclimate to the cold, as they would in nature. The pupae were then moved to 17 °C, and the number of successful eclosions were recorded. For 2 °C, a commercial fridge was used. Climate controlled rooms were used for the pre- and post-winter treatments. Stretching from August to April, the overwintering experiment corresponded well with the true length of the overwintering period, and temperatures correspond approximately to a mild winter in their native location (Figs. S1, S9)[54]. Although differences between *P. rapae* and *P. napi* overwintering location preference in nature remain unknown, the two species have similar preferences when developing directly[56]. Furthermore, they had similar pupation preferences when being reared under diapause-inducing conditions in the laboratory (most frequently along the top edges of the rearing cages).

## Field observations (citizen science data)

Observational data of adult *P. rapae* and *P. napi* from years 2010–2021 were gathered from the Swedish Species Information Centre[54]. Butterflies are conspicuous and often easy to identify to species level upon observation in the wild. Moreover, both *P. rapae* and *P. napi* are common in Sweden and relatively drab (for butterflies) making biases between them highly unlikely. Therefore, this study system is exceptionally suitable for citizen science data analyses.

Observations were grouped by year and province. Only the 13 provinces where both *P. rapae* and P. *napi* had been observed consistently across years were included in the analyses (see Fig. S1). The data set was cleaned by removing abnormally early or late observations (before Julian day 90, or after Julian day 270). Density curves of observations over time were produced for each unique combination of province and year using kernel density estimation (14-day bandwidth).

The date of the flight peaks—when observations of adult butterflies are most common—were estimated using numerical approximation (of $x$ where $f'(x) = 0$). Both *P. rapae* and *P. napi* are bivoltine in the sampled geographical range, and dates of the two peaks could be reliably extracted for 239 of 312 unique combinations of province and year. To generate a proxy metric for peak butterfly abundances that could be compared across years and provinces, the density at each flight peak was multiplied by the total number of observations for that year and province. The final data used for the analyses were generated from 5712 observations of *P. rapae* and 20,040 observations of *P. napi*.

## Software

All modelling was performed using Bayesian methods in Stan[137], through the package *brms*[138] in R (version 4.1.3)[139]. Other R packages used were *tidyverse*[140], *lubridate*[141] and *bayestestR*[142].

## Modelling

The effect of microclimate temperature on oviposition choice was modelled using a logistic regression, with species as the response variable, and average daily mean temperature as the predictor variable. For both translocation events, average daily mean temperature in each microhabitat were calculated by averaging the daily mean temperatures between 0800 and 1800 h. The time interval was chosen to represent the actual temperatures female butterflies experience and sample, and was based on when flying butterflies were observed in the field during the experiment. We tried to account for confounding

effects of other, site-specific, factors that could influence female choice (e.g., vicinity to nectar plants or wind exposure) by modelling microhabitat site as a group-level effect with random intercepts.

Temperature-dependent trait performance in the laboratory was estimated for three growth season traits: development rate, larval growth rate, and pupal mass retention. Development rate represents how quickly an individual develops, and was calculated as

$$\text{rate}_{\text{development}} = \frac{1}{\text{time}_{\text{development}}} \quad (1)$$

where development time is the time in days it took to complete the given life stage. Larval growth rate was calculated as

$$\text{rate}_{\text{growth}} = \frac{\text{mass}_{\text{pupation}}}{\text{time}_{\text{larval development}}} \quad (2)$$

representing the daily rate of mass gain in milligrams in the larval stage. Pupal mass retention was calculated as

$$\text{retention} = 1 - \frac{\text{mass}_{\text{pupation}} - \text{mass}_{\text{eclosion}}}{\text{mass}_{\text{pupation}}} \quad (3)$$

representing the proportion of the initial pupal mass that is retained as an adult.

The empirically supported Lobry–Rosso–Flandrois (LRF) function was used to fit nonlinear thermal performance curves (TPCs) to the growth and development rate data (Eq. 4; Appendix S1)[143]:

$$\text{rate} = R_{\text{opt}} \times \frac{(T - T_{\text{max}}) \times (T - T_{\text{min}})^2}{(T_{\text{opt}} - T_{\text{min}}) \times \left\{ (T_{\text{opt}} - T_{\text{min}}) \times (T - T_{\text{opt}}) - (T_{\text{opt}} - T_{\text{max}}) \times (T_{\text{opt}} + T_{\text{min}} - 2T) \right\}} \quad (4)$$

where $T$ is the current temperature, $T_{\text{min}}$, $T_{\text{opt}}$, and $T_{\text{max}}$ are the minimum, optimum, and maximum temperatures for performance, and $R_{\text{opt}}$ is the rate at $T_{\text{opt}}$. The LRF function has previously performed well when predicting *P. napi* development rates under naturally fluctuating thermal conditions, showing that its parameters are relevant not only under laboratory conditions, but also for natural processes[60]. The function has several desirable statistical properties, among them a good fit to empirical data from several insect species, and biologically meaningful parameters (Eq. (4))[144]. The latter is valuable for specifying informative priors, restricting nonsensical parameter values and combinations (e.g. $T_{\text{opt}} = 0\,°C$ and $T_{\text{min}} > T_{\text{max}}$), allowing for better specified models that can handle, for example, random family effects, rearing batch effects, sex effects and interactions among them. The LRF function was modified using conditional statements so that it evaluates to zero when temperatures are below $T_{\text{min}}$, or above $T_{\text{max}}$, thus being sensical at all temperatures, in turn improving the validity of the model fitting procedure. This methodology allows fitting TPCs without compromising the inclusion of important covariates, fixed factors and random group-effects, which is difficult in traditional nonlinear least-squares approaches.

The two species were compared using separate models for each life stage. Random variance in the data from each unique combination of life stage and species was homogenous on the logarithmic scale. Therefore, the LRF function was logarithmized and fitted using a lognormal error distribution. This simplified the modelling of sex, year, and random group-level effects, which were added as separate terms operating on the logarithmic scale. In each model, residual variance was allowed to vary with species. Container (larvae and ontogeny treatments) and family were modelled as group-level effects with random intercepts. Family intercepts were allowed to vary among temperature treatments, accounting for potential family differences in the shape of the TPC, and a separate family-variance

component was estimated for each species, since genetic effects might differ between them. Sex was dummy coded as −0.5, 0.5, and 0 for females, males and non-determined individuals, respectively (a 50/50 sex ratio is expected), and modelled as a slope effect with an intercept of 0 on the logarithmic scale. Since only the *P. napi* experiments were carried out over two years, year was also dummy coded as −0.5, and 0.5 for the *P. napi* data, and as 0 for the *P. rapae* data, and modelled in the same way as sex. The sex and year effects essentially average the TPC over the years and sexes (while also allowing the inclusion for non-sexed individuals in the model), and their parameter estimates represent the difference between the sexes and years. Pupal mass retention was modelled as a linear function of temperature (on the logarithmic scale), and the sex effect was allowed to vary with temperature, but other variables were modelled as previously described.

Temperature effects on survival were modelled using a logistic regression with temperature treatment specific intercepts. Container, family, and sex effects were modelled using the aforementioned approach. Since survival data is often noisy, with relatively strong batch effects, life stage was modelled as a group-level effect with random intercepts that were allowed to vary among temperature treatments. Although life stage is technically not a random variable, each life stage within a temperature treatment can be considered a batch of its own, and the distribution of estimated random intercepts conformed well with a Gaussian distribution (Fig. S3). Therefore, this approach gives a robust between-species comparison of survival at different temperatures. Species differences in winter survival were modelled as an intercept effect in a logistic regression, with sex and family modelled as previously described.

To explore cross-seasonal population dynamics in the citizen science data, two models were used: a 'growth season model' modelling butterfly abundances in the fall peak (of a given province and year) as a function of those in the previous spring peak, and an 'overwintering model' modelling butterfly abundances in the spring peak (of a given province and year) as a function of those in the fall peak the previous year. Flight peak abundance data appeared linear and homoscedastic when logarithmized, and were therefore transformed both as predictor and response. To describe the relationships between the flight peaks, a linear function with species-specific intercepts and slopes was fitted to the growth season data, and a linear function with only species-specific intercepts was fitted to the overwintering data, since it did not support the interaction term. A Gaussian error distribution was used, and province and year were modelled as group-level effects with random intercepts to account for confounding effects (e.g. a province having a particularly high number of butterfly-enthusiasts, or a certain year being particularly beneficial—for butterflies or butterfly-enthusiasts alike).

Finally, a Poisson regression was used to model yearly changes in number of butterfly observations between the years 2010–2021. For each species, the total number of observations within a year and province was calculated. The slope of the effect of year on total number of observations was estimated, and province was modelled as a as group-level effects with random intercepts.

All models were run using four parallel Markov chains, each running for 4000 iterations (first 2000 discarded as burn-in). Posterior modes were used for point estimates, and 90% credible intervals ($CI_{90}$) were used for uncertainties. Posterior predictive checks revealed that all models could successfully approximate the distributions of observed data, indicating good fits (Fig. S4). In most models, default priors in the brms package were used. In the nonlinear models, custom informative priors were used. Priors were identical for both species, and were as such unbiased. Any substantial species differences in the posteriors result from experimental data alone. For details, see Appendix S1 and Supplementary Data 1.

**Reporting summary**

Further information on research design is available in the Nature Portfolio Reporting Summary linked to this article.

## Data availability

All data required to reproduce the results in this study are available in the Figshare repository [https://doi.org/10.6084/m9.figshare.22657069][145].

## Code availability

All code required to reproduce the results in this study is available in the Figshare repository [https://doi.org/10.6084/m9.figshare.22657069][145].

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

## Acknowledgements
This study was made possible through funding from the Bolin Centre for Climate Research, and is a part of the Bolin Centre Research Area 8, research in biodiversity and climate. P.L. thanks the Research Council Formas (2017-00965) and VR (2017-04159, 2022-03343) for additional support. K.G. thanks VR (2021-04258) for additional support. We thank Olle Lindestad for making the butterfly illustrations used in Figs. 1 and 6. We thank Christer Wiklund for insightful comments on the ecology of *Pieris rapae* and *P. napi*. Last, but not least, we thank all the people that reported butterfly observations to Artportalen for their valuable contribution to the scientific community.

## Author contributions
L.v.S., P.L. and K.G. designed the study. P.C. and L.v.S. conducted the field experiments. L.v.S., K.H.G. and P.L. conducted the laboratory experiments. L.v.S. gathered the citizen science data and performed all analyses. L.v.S. drafted the manuscript. P.L. and K.G. contributed substantially to the structure and content of the manuscript, from the start to the final version. All authors approved the final manuscript.

## Funding

## Competing interests
The authors declare no competing interests.
