## [Peer review file · Nature Communications]

REVIEWER COMMENTS

Reviewer #1 (Remarks to the Author):

This study provided a very nice demonstration of how differences in thermal responses between two sympatric, congeneric butterfly species result in different seasonal patterns of abundance. The integration of lab and field experiments, modeling and field science data is powerful. The study clearly demonstrates contrasting patterns of summer (growing season) and overwintering (diapause) success between the two species, which are interpreted in terms of tradeoffs and seasonal specialization. The Bayesian modeling framework used in the analyses was a bit challenging to follow in spots, but has some real strengths: it allows them to use priors that reflect biological constraints on the TPCs, and to estimate mean curves while accounting for variables of lesser direct interest. Overall this is a very interesting and valuable contribution to our understanding of how closely related species respond and adapt to seasonal climatic variation near their species range boundaries. Despite this enthusiasm, we have several general concerns and specific suggestions for the authors to consider.

The Introduction presents a very strong argument for including seasonality as a niche axis, but does so without citing previous literature on the topic. The general idea of seasonal niches and specialization is not new: for example, there is a large literature on flowering phenology and seasonal partitioning of pollinators. Similarly, how differences in thermal niches are associated with differences in seasonal life cycle has been widely discussed. The current study should be more clearly connected to this broader literature. In addition, the Introduction does not adequately motivate the specific experiments and elements of the study and explain how they are linked together to answer their research questions of interest. This could be strengthened by expanding lines 88-96 to include more clear delineation of how the studies fit together, and foreshadowing what will be talked about in the discussion.

The Introduction and Discussion miss some opportunities to connect this study to previous work in these two species that is directly relevant to the story. The habitat differences between *rapae* and *napi* (and other species of the *napi* complex in North America) are well known: *rapae* prefer open areas, *napi* and *oleracea* prefer more shady areas and forest edges (cf Chew 1981), and the range of *rapae* extends into warmer and drier regions. *P. rapae* in North America and Australia achieve high rates of survival, growth and development at temperatures above 30 C (Jones and Ives 1979, Kingsolver 2000). Unlike *P. napi*, *P. rapae* is highly dispersive with a high maximum intrinsic rate of population increase, and high gene flow that may limit local adaptation. Placing the study in the larger context of these species outside of Sweden will strengthen the paper.

The focus on seasonal tradeoffs between summer growth and overwintering survival seems overemphasized. Is this really different from warm (*rapae*) vs cold (*napi*) adapted species, which have been widely described in other systems? *P. rapae* is highly dispersive and near its northern range limit in Sweden, so gene flow from larger, southern populations could prevent local adaptation (while maintaining genetic variation). Also overwintering survival of *rapae*, while lower than *napi*, is still quite high given the long overwinter period. More generally, the tradeoffs component of the Discussion— in particular the first 6 paragraphs— could be substantially reduced, including the discussions of standing genetic variation and gene expression that seem tangential to the main points of the study.

Line Edits

Line 60-61: Needs to be explained further if included - we are not familiar with salmon!

Line 77: Are these dynamics boom-bust? Seems incorrect to use this language to describe predictable seasonal variation driven by weather

Lines 99-100: What is the range of microhabitat? How were these selected?

Line 101: Transplant might not be the correct word to use here if they're in pots

Line 116: should say 24-hr day lengths

Lines 140-142: This is too opaque – more explanation needed for comprehension

Lines 144-148: The egg and larval data for *napi* are previously published. This should also be noted in the associated figure legends. Are the microclimate data also from a previous paper (Greiser et al

2022)? What is the relationship between these papers and the current study?

Lines 156-157: Do the experimental temperatures also correspond to field temperatures? Justify the selected temperatures.

Lines 183: What is a batch?

Fig 2: Is the CI shaded based on temp or species or another factor?

Fig 3: Pretty clear - but it's not just that the curve is "right-shifted" - it's that *P. rapae* have higher overall growth rates, so they're mostly performing better at the "optimal" temperatures for both species

Line 297-298: Are these two developed *P. rapae* really that important statistically and biologically, to devote this and a whole discussion paragraph to talking about them?

Fig 5: The conclusions about density-dependence are too big a jump for the readers: Lines 302-303 and 312-313 need more explicit explanation for how the 1:1 line relates to exponential population growth, thus a shallower slope is likely due to density-dependence.

Line 324-326: It is not clear that these are fitness 'costs', just that *rapae* has both higher population growth in summer and lower overwinter survival than *napi*

Lines 357-372: Reduce discussion of tradeoffs. Many factors can maintain genetic variance other than genetic tradeoffs (gene flow is more likely here).

Line 364: This claim is speculative - selection was not estimated in this study

Line 366-367: Add how you are figuring this out: differences in survival across families

Lines 373-383: Omit this- speculative, not well connected to the actual results, and only information for *napi*.

Line 380-381: What is this referencing? Unclear as to meaning.

Lines 388-401: Reduce- again, plausible but speculative.

Lines 402-407: Could be clearer: expand on habitat differences in introduction to add relevance to this section

Line 412: Again, boom-bust language doesn't fit here

Reviewer #2 (Remarks to the Author):

The authors of this manuscript revealed divergent patterns of seasonal population dynamics in two sympatric butterfly species by analyzing citizen science data. Using a combination of experiments in the field and laboratory, the authors further attributed these divergent patterns to the divergent thermal performance and overwintering success of the two species. The study is quite interesting to me and should of interest to the wide readership of Nature Communications. The conclusions drawn in this manuscript are based on detailed experimental data and long-term field observation. Here, I proposed several major comments as well as some line-by-lines hoping to help improve the manuscript.

Major comments:

1. The authors mentioned in Line 84 that *P. rapae* and *P. napi* differ in their host plant use in nature. However, in the oviposition choice experiment, only *Brassica napus napus* was used to test the microhabitat choice of butterfly species. Do the two species like this plant species equally? If not, what if how much species A like and species B dislike the plant depends on the temperature (for example, the preference is induced by a temperature-sensitive chemical signal)? I think the authors should better explain their use of just one host plant species here.

2. The authors used air temperatures measured by shaded loggers placed next to the host plants to represent temperatures experienced by the ovipositing butterflies and developing eggs, larvae and pupae. However, the "body temperature" of these individuals were determined by both a combination of environmental variables such as air temperature, radiation, wind speed, leaf temperature and the individual' behavioral thermoregulation such as changing angle towards the sun. Ideally, the authors

would have directly measured the body temperatures of adult butterflies as well as the temperatures of the eggs, larvae and pupae. Given this might be challenging, the author may consider comparing the temperatures measured by the shaded loggers with body temperatures of adult butterflies (measured using thermal couple or thermal camera) and the leaf temperatures using a relatively small sample size.

3. The authors investigated the temperature dependence of development rates for each separate life stage, which was very interesting. However, they did not discuss much about the differences between those TPCs for different life stages (say, the T_{max} is higher for eggs than for Larvae and Pupae). Readers may want to know about reasons why the TPCs differ between life stages, the ecological implications and the implications for the scientific questions addressed by this study.

Line-by-lines:

Line 68: It's not quite clear here what kind of idea the authors are testing. Consider raising a detailed scientific question here or clearly describing the hypothesis/theory the authors are going to test.

Line 95-96: This statement assumes that overwintering survival is a kind of performance, which I could not find support for. Consider changing it to something like "trade-offs between thermal performance in the growth season and overwintering survival".

Line 95-96: It was not clear that whether the two species diapause in similar microclimates or not, which may explain why the difference between overwintering survivals of the two species was larger in the field than in the lab.

Line 186-187: Please indicate what "other environmental variables" were considered here.

Line 319-320: I don't think this is a valid statement. The authors did not use separate data (such as field survey data collected by scientists) to test the accuracy of population dynamics based on citizen science data.

Line 348-350: Are microhabitats that are relatively warmer when butterflies lay eggs also warmer later when eggs develop and larvae grow? Consider showing the whole temperature profiles (cover all stages) of the air temperature measured by shaded loggers for various microhabitats in supplementary materials.

Line 362: Consider deleting "and drier" as the authors did not consider humidity in this study.

Figure 1: What do the heights of the curves mean? Consider adding the y-axis.

Dear Reviewers,

Thank you for thoroughly reviewing our work and for all your valuable comments. They have led to clear improvements of the manuscript, notably making it more coherent and streamlined, and less speculative about tangential topics. Additionally, we have expanded the data on climatic conditions in the supplementary material, which we believe has strengthened the results further.

The format of the revised version is adapted to that of *Nature Communications* (e.g., materials and methods last). Additionally, We have now also annotated all the scripts required for reproducing the results and the figure components. Seeds have been specified for full reproducibility, and point estimates, CI's, and figures have been updated to match the exact results obtained by running the script (owing to the stochasticity of the Monte Carlo sampling – the general results remain unchanged). We have attached the scripts and data (“Data and scripts von Schmalensee et al. 2023.zip”) that will be uploaded to the Dryad repository, should our manuscript be accepted for publication.

To make your revisions easier, we have attached a version of the revised manuscript where changes based on your comments have been marked **red** (file labeled “MARKED_” before the rest of the file name), together with the unmarked version. Below, we address your general and specific comments, and our replies have been *italicized* for clarity.

Thank you again for your time and effort.

Reviewer 1 (Remarks to the Author):

This study provided a very nice demonstration of how differences in thermal responses between two sympatric, congeneric butterfly species result in different seasonal patterns of abundance. The integration of lab and field experiments, modeling and citizen science data is powerful. The study clearly demonstrates contrasting patterns of summer (growing season) and overwintering (diapause) success between the two species, which are interpreted in terms of tradeoffs and seasonal specialization. The Bayesian modeling framework used in the analyses was a bit challenging to follow in spots, but has some real strengths: it allows them to use priors that reflect biological constraints on the TPCs, and to estimate mean curves while accounting for variables of lesser direct interest. Overall this is a very interesting and valuable contribution to our understanding of how closely related species respond and adapt to seasonal climatic variation near their species range boundaries. Despite this enthusiasm, we have several general concerns and specific suggestions for the authors to consider.

The Introduction presents a very strong argument for including seasonality as a niche axis, but does so without citing previous literature on the topic. The general idea of seasonal niches and specialization is not new: for example, there is a large literature on flowering phenology and seasonal partitioning of pollinators. Similarly, how differences in thermal niches are associated with differences in seasonal life cycle has been widely discussed. The current study should be more clearly connected to this broader literature. In addition, the Introduction does not adequately motivate the specific experiments and elements of the study and explain how they are linked together to answer their research questions of interest. This could be strengthened by expanding lines 88-96 to include more clear delineation of how the studies fit together, and foreshadowing what will be talked about in the discussion.

The Introduction and Discussion miss some opportunities to connect this study to previous work in these two species that is directly relevant to the story. The habitat differences between *rapae* and *napi* (and other species of the *napi* complex in North America) are well known: *rapae* prefer open areas, *napi* and *oleracea* prefer more shady areas and forest edges (cf Chew 1981), and the range of *rapae* extends into warmer and drier regions. *P. rapae* in North America and Australia achieve high rates of survival, growth and development at temperatures above 30 C (Jones and Ives 1979, Kingsolver 2000). Unlike *P. napi*, *P. rapae* is highly dispersive with a high maximum intrinsic rate of population increase, and high gene flow that may limit local adaptation. Placing the study in the larger context of these species outside of Sweden will strengthen the paper.

The focus on seasonal tradeoffs between summer growth and overwintering survival seems overemphasized. Is this really different from warm (*rapae*) vs cold (*napi*) adapted species, which have been widely described in other systems? *P. rapae* is highly dispersive and near its northern range limit in Sweden, so gene flow from larger, southern populations could prevent local adaptation (while maintaining genetic variation). Also overwintering survival of *rapae*, while lower than *napi*, is still quite high given the long overwinter period. More generally, the tradeoffs component of the Discussion— in particular the first 6 paragraphs— could be substantially reduced, including the discussions of standing genetic variation and gene expression that seem tangential to the main points of the study.

We generally agree with your points. First, we have added the suggested references. Moreover, a new study by Vives-Inglá et al. (2023) in Ecological Monographs (published during this review process)

investigates time-dependent mortality and microhabitat use in Spanish populations of P. napi and P. rapae. These studies are now cited accordingly (lines 95-97, 99).

As suggested, we have expanded the background on seasonality in niche theory (lines 65-75). Like you mention, there is a large body of literature on the role of seasonality in niche partitioning, and we now refer to some empirical examples of shifting flowering phenologies (as proposed), and host-shift mediated phenology changes in Rhagoletis flies. We also refer to more of the theoretical work that has been done on coexistence in shared (seasonally) variable environments. However, as we mention in this new section, we are less aware of empirical work investigating seasonal specialization (i.e., differences in 'season-specific fitness') in synchronous organism groups with longer, non-overlapping, generations as the butterflies studied here (cf. plankton). If there are references that we have overlooked, we apologize and welcome suggestions.

We agree with the point that we cannot confidently infer trade-offs as driving the seasonal differences given our experimental setup. Although we think the idea of cross-seasonal trade-offs is interesting and important to bring up, we have now substantially reduced the focus on this subject in the discussion, and instead emphasized other potential causes (e.g. the point you make about gene swamping from southern populations; lines 253-256).

Last, thank you for highlighting our TPC-fitting approach. We believe that these methods could be highly beneficial for future studies, e.g. on local thermal adaptation, and we are glad their utility came across in the manuscript.

Line Edits

Line 60-61: Needs to be explained further if included - we are not familiar with salmon!

Fair point. We have expanded on this slightly to drive the point that seasons can be thought of as relatively coarse-grained 'locations' on a temporal axis (lines 56-59)

Line 77: Are these dynamics boom-bust? Seems incorrect to use this language to describe predictable seasonal variation driven by weather.

You are right, 'boom-bust' (in the density dependent/feedback sense) is not correct here, since the pattern is caused by extrinsic factors. We have removed the expression 'boom-bust' from the entirety of the manuscript.

Lines 99-100: What is the range of microhabitat? How were these selected?

The microhabitats were selected based on the temperature distributions of 110 sites (Greiser et al. 2022) during the previous summer. The sites were selected manually (to ensure accessibility during the experiment) with the aim of capturing as much of the variation in mean temperatures, and variation around those mean temperatures, as possible. This has now been clarified in the Methods section (lines 360-364)

Line 101: Transplant might not be the correct word to use here if they're in pots

Thank you for pointing this out. Changed to 'translocated' throughout the manuscript.

Line 116: should say 24-hr day lengths

*We meant to describe that light conditions were 23 hours light/1 hour dark each day. We have now changed the wording to clarify this (**line 381**).*

Lines 140-142: This is too opaque – more explanation needed for comprehension

*We see that the original text was a bit unclear. What we meant to say was that, since not a single *Pieris rapae* pupa survived in the 40°C treatment, any survivor that was recorded as a result of an increased sample size would just further strengthen the conclusion that *P. rapae* tolerate heat better than *P. napi*. As such, the general conclusions about heat tolerance differences are robust, despite the low powered pupal treatment. We have changed the wording slightly, and we hope it is clearer now (**lines 406-409**)*

Lines 144-148: The egg and larval data for *napi* are previously published. This should also be noted in the associated figure legends. Are the microclimate data also from a previous paper (Greiser et al 2022)? What is the relationship between these papers and the current study?

*Thank you for pointing this out. We have changed the legend for Fig. 3 accordingly. And you are correct – daily mean temperatures are based on microclimate data that is now published in Greiser et al. 2022 (well spotted!). We now reference this accordingly to point out the relationship between the papers (**lines 373-374, 863-864, 871-872**).*

Lines 156-157: Do the experimental temperatures also correspond to field temperatures? Justify the selected temperatures.

*The experimental overwintering temperatures correspond approximately to a mild winter, and have therefore been used to successfully overwinter pupae several times in our laboratory. We clarify how the temperatures relate to natural conditions in the Methods section. We also added a supplementary figure where weather station temperatures (from the same region as the butterfly populations) are plotted over the experimental overwintering temperatures (Fig. S9, see below). We reference this in the Methods section (**lines 424-426**). We have also emphasized the fact that our winter treatment lacks extreme cold or warm winter periods (and is therefore likely relatively benign) in the Discussion section (**lines 288-290**).*

Lines 183: What is a batch?

We meant to use ‘batch’ as defined previously in the Methods section (lines 368-371), but have now changed the wording for clarity: ‘For both translocation events, average daily mean temperature in each microhabitat were calculated by ...’ (lines 453-455)

Fig 2: Is the CI shaded based on temp or species or another factor?

The CI is colored by species probability. We have now clarified this in the figure legend (line 863). Thanks for pointing it out.

Fig 3: Pretty clear - but it’s not just that the curve is “right-shifted” – it’s that *P. rapae* have higher overall growth rates, so they’re mostly performing better at the “optimal” temperatures for both species

Indeed (our results to some extent support the ‘warmer is better’ hypothesis). We now point this out in the Results section (lines 147-148).

Line 297-298: Are these two developed *P. rapae* really that important statistically and biologically, to devote this and a whole discussion paragraph to talking about them?

We see the point, and we have therefore substantially reduced the discussion of this finding. However, we actually do believe that they are indicative of a qualitative, mechanistic, difference in diapause time-keeping, and as such a finding of significant importance. In fact, preliminary data from an ongoing study (inspired by our findings here) led by Christer Wiklund support our hypothesis. We hope that this can justify the little speculation that remains (lines 285-288).

Fig 5: The conclusions about density-dependence are too big a jump for the readers: Lines 302-303 and 312-313 need more explicit explanation for how the 1:1 line relates to exponential population growth, thus a shallower slope is likely due to density-dependence.

We now explain that a shallower slope represents that each additional individual in one generation contributes less to the size of the next generations (lines 188-189)

Line 324-326: It is not clear that these are fitness 'costs', just that rapae has both higher population growth in summer and lower overwinter survival than napi

*We agree that the use of 'costs' here is confusing. Instead, we describe how much higher average fitness (i.e. the average contribution to the next reproducing generation) *P. napi* have compared with *P. rapae* over the winter period (5 times higher) (lines 212-214).*

Lines 357-372: Reduce discussion of tradeoffs. Many factors can maintain genetic variance other than genetic tradeoffs (gene flow is more likely here).

Based on your comments, we have substantially reduced the focus on trade-off in the new manuscript version (and we agree that the detailed speculation was unwarranted). However, we think it is still valuable to mention trade-offs as one potential mechanism (among others, like gene flow) that could cause patterns of 'seasonal specialization', and why. We also believe it should be mentioned that we find some evidence of genetic variation for overwintering success in the laboratory, which together with high winter mortality in nature can be taken to imply that natural selection should favor better overwinterers (lines 264-266).

Line 364: This claim is speculative – selection was not estimated in this study

Fair point. We removed the claim that there is strong selection (lines 264-266).

Line 366-367: Add how you are figuring this out: differences in survival across families

Since we reduced the focus on trade-offs, we omitted this explanation entirely, and instead refer to the estimated parameters in Table S1 (from which this can be derived). What we did originally was to translate the variance component (estimated for among-family variation, represented as the standard deviation) into probabilities. The model is linear in 'logit-space', but on the scale we are interested in (i.e., probabilities) the standard deviation that describes the variance component is difficult to interpret – it changes depending on what value it is centered on. Thus, to provide a tangible representation of the among-family variance we represented the standard deviation (i.e., the average family-specific deviation from the mean overwintering success) in probabilities, given that the average probability was 50% (which makes the interval symmetric).

Lines 373-383: Omit this– speculative, not well connected to the actual results, and only information for napi.

We have now substantially reduced the detailed speculation about trade-offs. We hope that our changes are satisfactory.

Line 380-381: What is this referencing? Unclear as to meaning.

This sentence has been omitted, but it referenced descriptions of genetically encoded production of (costly) compounds used for coping with temperature stress. The reasoning was this: a species with a (genetically determined) higher baseline production of, say, costly heat shock proteins as an adaptation to extreme seasons might fare worse during seasons with benign temperatures than would a similar species without that adaptation. Such allocation trade-offs could help maintaining 'seasonal specialization'. It is clear that our reasoning was obscure given the original sentence.

Lines 388-401: Reduce— again, plausible but speculative.

*We have reduced this section (**lines 285-288**)*

Lines 402-407: Could be clearer: expand on habitat differences in introduction to add relevance to this section

*Agreed. We have now made this part a bit clearer in the discussion (**lines 293-298**), and we mention *Pieris rapae*'s preference for agricultural fields in the introduction (**lines 94-95**).*

Line 412: Again, boom-bust language doesn't fit here

The term 'boom-bust' has been removed entirely from the manuscript.

Reviewer 2 (Remarks to the Author):

The authors of this manuscript revealed divergent patterns of seasonal population dynamics in two sympatric butterfly species by analyzing citizen science data. Using a combination of experiments in the field and laboratory, the authors further attributed these divergent patterns to the divergent thermal performance and overwintering success of the two species. The study is quite interesting to me and should be of interest to the wide readership of Nature Communications. The conclusions drawn in this manuscript are based on detailed experimental data and long-term field observation. Here, I proposed several major comments as well as some line-by-lines hoping to help improve the manuscript.

Major comments:

1. The authors mentioned in Line 84 that *P. rapae* and *P. napi* differ in their host plant use in nature. However, in the oviposition choice experiment, only *Brassica napus* was used to test the microhabitat choice of butterfly species. Do the two species like this plant species equally? If not, what if how much species A like and species B dislike the plant depends on the temperature (for example, the preference is induced by a temperature-sensitive chemical signal)? I think the authors should better explain their use of just one host plant species here.

This is an interesting point: is there an interaction between host plant and microclimate that is not represented as differences among host plants? Intuitively, this appears relatively unlikely. The ability of the butterflies to judge host plant or microclimate quality is beneficial in many conceivable scenarios, whereas the interaction between the two might not add much information (and as such is less likely to evolve). For example, picking a high-quality host plant of a certain species is likely informative of the microclimatic conditions in which it grows. Still, the usefulness for the butterflies in having an 'internal model' of the interaction between host plant and climate cannot be entirely ruled out (although such an internal model is more complex, and might be more difficult to evolve). For instance, it is possible that a certain plant of currently superb quality is likely soon wilting in certain microclimates (microclimates that in and of themselves might be benign for the butterfly), but not in others.

*As you note, our experiment cannot estimate such an effect, since we only used one host plant of standardized quality. We argue, however, that this is not crucial for our conclusions. Previous studies have shown microhabitat separation in wild *Pieris rapae* and *P. napi* (Friberg & Wiklund 2019, Vives-Inglá et al. 2023) where, on wild host plants, *P. rapae* prefer to lay eggs in warm/dry microhabitats. To expand these findings, we attempted to remove host plant quality from the equation by standardizing the quality of translocated plants, keeping them out for a maximum of 3 days, and watering them daily (added to **lines 366-367**). We did this to minimize environmentally induced differences among host plants (e.g. differences in chemical signals). Additionally, we used small *Brassica napus* plants with only cotyledon leaves (Fig. S2), which have been shown to be highly attractive to *P. napi* (likely because of their high concentration of glucosinolates; Friberg & Wiklund 2016). *Brassica napus* is also highly attractive to *P. rapae* – *P. rapae* is a well-known pest species on *B. napus* (CABI 2022 – now cited at **line 366**). With all evidence taken together, we argue that it is highly unlikely that we would get qualitatively results if we used another, standardized, host plant in an otherwise equivalent experiment. We hope that you find our arguments reasonable and convincing.*

2. The authors used air temperatures measured by shaded loggers placed next to the host plants to represent temperatures experienced by the ovipositing butterflies and developing eggs, larvae and pupae. However, the “body temperature” of these individuals were determined by both a combination of environmental variables such as air temperature, radiation, wind speed, leaf temperature and the individual’ behavioral thermoregulation such as changing angle towards the sun. Ideally, the authors would have directly measured the body temperatures of adult butterflies as well as the temperatures of the eggs, larvae and pupae. Given this might be challenging, the author may consider comparing the temperatures measured by the shaded loggers with body temperatures of adult butterflies (measured using thermal couple or thermal camera) and the leaf temperatures using a relatively small sample size.

*This is also a good point. We emphasize that we did not intend to claim that internal body temperatures of ovipositing females are necessarily driving their choices (we have now changed the word ‘driver’ to ‘predictor’ at line 126 – we apologize for the confusion). Rather, we are interested in the fact that their oviposition choices – regardless of what they are based on – correlate with microclimatic conditions. This is particularly relevant since we know from a previous study on *P. napi* (von Schmalensee et al. 2021 – the study was carried out in the exact same area as the oviposition experiment) that these natural microclimatic conditions directly influence egg and larval performance in a predictable way (and the findings likely extend to *P. rapae* as well). That is why females’ choices influencing the thermal regimes experienced by the offspring is important. We hope we can convince you that, although interesting, taking thermocouple measurements of adult butterflies (which requires waiting until July) would not add very much to the present story regardless of the outcome of those measurements (but potentially a better mechanistic understanding of how the choices are made).*

To highlight how egg-laying choices (however they are made) reflect the thermal environments of the offspring, we have now added two figures to the Supplementary Information (Fig. S7 and Fig. S8, shown in the next page). The first figure shows how thermal regimes experienced by hypothetical offspring during the month after oviposition depends on oviposition temperature (the temperature loggers remained in the field after the oviposition experiment). Granted, actual body temperatures might differ from these measurements (e.g. due to thermoregulation), but we emphasize that the host plants are small and close-ground, and that these shaded loggers have previously been used for successful performance predictions (von Schmalensee et al. 2021).

*The second figure expands these results, and show how microhabitat separation might reflect competitive differences between *P. rapae* and *P. napi*. We estimated these differences by first calculating average larval development and growth rates for each microclimate (Appendix S1). This was done using rate summation (as in von Schmalensee et al. 2021), where performance is calculated by integrating the temperature and the temperature-dependent performance at each time point (here hourly). Then, we calculated the ratio of *P. rapae* to *P. napi* performance, and overlaid the scaled distribution of performance differences on the oviposition preference curve. This revealed a clear relationship between predicted *P. rapae*/*P. napi* performance and the estimated probability of an egg being *P. rapae*!*

Finally, we would like to point out that we agree with you on the general importance of accurately capturing operational temperatures in nature, and we are aware that the mechanisms that govern those temperatures are complex and can operate at extremely small scales. For quantitative predictive models of ectotherm performance in nature, these considerations are crucial.

Female choice predicts future temperatures

The figures show the relationship between mean daily oviposition temperature and the temperature of the same site the following month.

Oviposition in cold  Oviposition in warmth

3. The authors investigated the temperature dependence of development rates for each separate life stage, which was very interesting. However, they did not discuss much about the differences between those TPCs for different life stages (say, the T_{max} is higher for eggs than for Larvae and Pupae). Readers may want to know about reasons why the TPCs differ between life stages, the ecological implications and the implications for the scientific questions addressed by this study.

*Again, we agree that these questions are very interesting. How does temperature dependence vary among life stages? How does this reflect the ecology and life history of the species (e.g., sessile vs. motile life stages)? However, as it stands, we believe that an elaboration on, and discussion of, these topics would distract from the main story and make the manuscript (and reference list – there is a whole body of literature related to this topic) substantially longer. Instead, we encourage using our results (presented in detail in Tab. S1), or our raw data (which will be made available should the manuscript be accepted) to investigate exactly this. (Models should probably be tailored for that particular research question – e.g. being species specific rather than life-stage specific.) We have now emphasized this in the results section (**lines 233-235**). This comment relates to your previous comment about operative body temperatures: in order to truly understand the ecological implications of life stage differences in TPCs, one needs to study the organism closely (e.g. – do the larvae thermoregulate, do their feeding on leaves induce cooling processes, etc?). As such, we hope that you agree that this topic deserves a study of its own, and that the current manuscript is not the best fit for it!*

Line-by-lines:

Line 68: It's not quite clear here what kind of idea the authors are testing. Consider raising a detailed scientific question here or clearly describing the hypothesis/theory the authors are going to test.

*Thank you for pointing this out. We now clarify this (**line 76**), and explicitly state our hypothesis in the Introduction, linking it to our methodology (**lines 101-104**)*

Line 95-96: This statement assumes that overwintering survival is a kind of performance, which I could not find support for. Consider changing it to something like “trade-offs between thermal performance in the growth season and overwintering survival”.

*Fair point. We have replaced this particular sentence (**lines 112-115**), and others that implicitly referred to overwintering survival as a performance trait.*

Line 95-96: It was not clear that whether the two species diapause in similar microclimates or not, which may explain why the difference between overwintering survivals of the two species was larger in the field than in the lab.

*We currently do not know exactly where these species overwinter in nature. However, the two species pupate in the same manner (the pupae are morphologically very similar), they attach to a silk mat with a cremaster and a silk girdle, and during summer (while collecting host plants) we have observed both *P. rapae* and *P. napi* pupae in similar locations on both host plants and the surrounding environment. So, in nature, non-diapausers seem to have similar pupation site preferences in both species (see also CABI 2022).*

*Additionally, when reared in the laboratory, diapausing *P. rapae* and *P. napi* pupated in similar locations (most frequently along the upper edges in the rearing cages). So while we are not certain of their diapause locations in nature, evidence points to *P. rapae* and *P. napi* preferences being similar. We now elaborate on this in the Methods (lines 426-429). It should be mentioned that we believe the dominant reason for the lower overwintering success in nature is that natural winters are less benign than our experimental overwintering conditions, since wild individuals will experience more extreme conditions, both through warm and cold spells. We now show this in Fig. S9, and mention it in the Discussion (lines 288-290)*

Line 186-187: Please indicate what “other environmental variables” were considered here.

This ‘random effect’ attempts to account for confounding factors that influence how attractive a site is to ovipositing females, apart from temperature. Since we cultivated two batches of host plants and conducted two translocations per site – each with a unique average oviposition temperature – we can attempt to model this. It is difficult to determine the exact confounding factors that might influence how female butterflies perceive the microhabitats, but we can speculate that things like proximity to nectar plants, wind exposure, etc., all might influence a site’s attractiveness. We have now clarified this in the text (lines 457-459).

Line 319-320: I don't think this is a valid statement. The authors did not use separate data (such as field survey data collected by scientists) to test the accuracy of population dynamics based on citizen science data.

We agree. We have removed this sentence.

Line 348-350: Are microhabitats that are relatively warmer when butterflies lay eggs also warmer later when eggs develop and larvae grow? Consider showing the whole temperature profiles (cover all stages) of the air temperature measured by shaded loggers for various microhabitats in supplementary materials.

We have now done that (Fig. S7, and also Fig. S8 – see above). We thank you for this comment – these were valuable additions to the study that strengthened our conclusions.

Line 362: Consider deleting “and drier” as the authors did not consider humidity in this study.

Deleted!

Figure 1: What do the heights of the curves mean? Consider adding the y-axis.

At first, we reasoned that we could omit the axis for simplicity since the actual numbers do not matter (they are simply the number of weekly observations, not actual abundances). However, we now see that this in fact instead might make it harder to interpret the figure, and we have now changed the axes accordingly.

References

- CABI. Invasive Species Compendium. Wallingford, UK: CAB International. (2022).
<https://www.cabi.org/isc>.
- Friberg, M. & Wiklund, C. Butterflies and plants: preference/performance studies in relation to plant size and the use of intact plants vs. cuttings. *Entomol. Exp. Appl.* **160**, 201–208 (2016).
- Friberg, M. & Wiklund, C. Host preference variation cannot explain microhabitat differentiation among sympatric *Pieris napi* and *Pieris rapae* butterflies. *Ecol. Entomol.* **44**, 571–576 (2019).
- Greiser, C., von Schmalensee, L., Lindestad, O., Gotthard, K. & Lehmann, P. Microclimatic variation affects developmental phenology, synchrony and voltinism in an insect population. *Funct. Ecol.* **36**, 3036–3048 (2022).
- Maria Vives-Inгла *et al.* Interspecific differences in microhabitat use expose insects to contrasting thermal mortality. *Ecol. Monogr.* **(Early View)**, (2023).
- von Schmalensee, L., Hulda Gunnarsdóttir, K., Näslund, J., Gotthard, K. & Lehmann, P. Thermal performance under constant temperatures can accurately predict insect development times across naturally variable microclimates. *Ecol. Lett.* **24**, 1633–1645 (2021).

REVIEWERS' COMMENTS

Reviewer #1 (Remarks to the Author):

The authors have thoroughly addressed the questions and suggestions raised in my earlier review. This is a very interesting study and analysis and a well-written paper.

Reviewer #2 (Remarks to the Author):

I have carefully read the revised version again. The manuscript has been much improved. The author has provided a detailed response to each of my comments and eliminated my concerns mostly. I am happy to recommend this paper to be accepted as it is now.